# Which Position for Novice Surgeons? Effect of Supine and Prone Positions on Percutaneous Nephrolithotomy Learning Curve

**DOI:** 10.3390/medicina60081292

**Published:** 2024-08-10

**Authors:** Ender Cem Bulut, Uğur Aydın, Çağrı Coşkun, Serhat Çetin, Ali Ünsal, Fazlı Polat, Bora Küpeli

**Affiliations:** 1Urology Department, School of Medicine, Gazi University, Ankara 06560, Turkey; scetin86@yahoo.com (S.Ç.); aunsal@gmail.com (A.Ü.); fpolat@gazi.edu.tr (F.P.); borakupeli@yahoo.com (B.K.); 2Urology Department, Ağrı Training and Research Hospital, Ağrı 04200, Turkey; ugurr.aydinn@hotmail.com (U.A.); drcagricoskun@gmail.com (Ç.C.)

**Keywords:** percutaneous nephrolithotomy, prone position, supine position, urolithiasis

## Abstract

*Background and Objectives*: Percutaneous nephrolithotomy (PCNL) is a current treatment method with high success rates and low complication rates in treating large kidney stones. It can be conducted in different positions, especially supine and prone positions. PCNL in the supine position is becoming increasingly common due to its advantages, such as simultaneous retrograde intervention and better anesthesia management. This study aimed to assess how the choice of position impacts the PCNL learning curve. *Materials and Methods*: The results of the first 50 consecutive PCNL cases performed by two separate chief residents as primary surgeons in supine and prone positions in a reference center for stone treatment between August 2021 and January 2023 were evaluated. The two groups’ demographic and clinical data, stone-free rates, operation times, and fluoroscopy times were compared. *Results*: While the mean operation time was 94.6 ± 9.8 min in the supine PCNL group, it was 129.9 ± 20.3 min in the prone PCNL group (*p* < 0.001). Median fluoroscopy times in the supine PCNL and prone PCNL groups were 31 (10–89) seconds and 48 (23–156) seconds, respectively (*p* = 0.001). During the operation, the plateau was reached after the 10th case in the supine PCNL group, while it was reached after the 40th case in the prone PCNL group. *Conclusions*: For surgeons who are novices in performing PCNL, supine PCNL may offer both better results and a faster learning curve. Prospective and randomized studies can provide more robust conclusions on this subject.

## 1. Introduction

Percutaneous nephrolithotomy (PCNL) is a method that has been used in the treatment of large kidney stones since 1976 and is recommended as the first choice of treatment for kidney stones larger than 2 cm and those over 1.5 cm in the lower calyx [1,2]. The positions used in PCNL have historically undergone several changes and modifications. Traditionally performed in the prone position, surgeons began performing PCNL in the supine position in 1987 after Valdivia and his colleagues introduced this technique [3]. Respiratory failure due to increased thoracic pressure, difficulties experienced by the anesthetist in airway control, and the risk of ocular and neurological injury due to multiple pressure points in the prone position led to research on PCNL in the supine position [4,5,6,7]. In addition to providing advantages such as working with lower intrarenal pressure and spontaneous stone retrieval, the supine position offers simultaneous retrograde intervention [8]. Grosso et al. also defined a modified prone position for PCNL which makes simultaneous ureteroscopy more challenging [9].

This study aimed to evaluate the effect of two different positions (supine and prone) on the PCNL learning process by assessing the operational outcomes of two chief residents who performed PCNL for the first time as primary surgeons, each using a different position.

## 2. Materials and Methods

Data from consecutive prone and supine PCNL operations performed by two separate chief residents as primary surgeons under the supervision of a mentor at Gazi University, School of Medicine, Urology Department, between August 2021 and January 2023 were retrospectively examined. One of the residents performed the first 50 PCNL operations in the prone position and the other performed the other 50 in the supine position. In Turkey, general practitioners receive specialty training in training clinics after passing the ‘Turkish Medical Specialization Examination’ (TUS), and this period is defined as residency. After this training, doctors earn the right to become specialist doctors. Residents are defined as chief residents in the final year of the 5-year urology residency. In our clinic, during the residency training period, chief residents perform PCNL as the primary surgeon. Prior to this, chief residents do not perform PCNL as primary surgeons; however, they assist in and observe PCNL cases extensively. Before the chief residency period, their tasks in the operation include placing a 5 Fr ureteral catheter and assisting the chief resident with equipment transfer. A mentor is present in the operating room in each case.

For each PCNL case, we recorded age, gender, body mass index (BMI), side, stone location, stone size, Hounsfield unit (HU), stone-free rate, operation time, fluoroscopy time, and complications. Preoperative stone size was evaluated by calculating the largest diameter using computed tomography (CT). If more than one stone was found, the size was calculated by adding up the dimensions. Stone-free was defined as the absence of residues or fragments less than 4 mm. The fifty consecutive PCNL operations for both supine and prone positions were divided into five groups of ten cases each.

All patients were adults (≥18 yo). Patients who underwent endoscopic combined intrarenal surgery (ECIRS) were excluded from this study. The condition of not having undergone surgery before was not required for inclusion in this study. The operations were performed after obtaining a sterile urine culture. To achieve a sterile urine culture, patients with a positive preoperative urine culture received antibiotic treatment for 7–10 days based on the antibiogram results, either as outpatients or inpatients. Operation time was calculated from anesthesia induction to double-J (DJ) stent placement. In supine PCNL cases, a 5 Fr ureteral catheter was positioned on the stone side in the Galdakao-modified Valdivia position, and the operation proceeded from this position. For prone PCNL, the same urethral catheter was placed in the lithotomy position, and then, the patient was repositioned to the prone position. Access to the calyx was achieved using an 18 G/20 cm percutaneous access needle with fluoroscopy, and a 0.035-inch hydrophilic guide wire was passed through the needle. Access was obtained from the most suitable calyx to ensure adequate lithotripsy and stone extraction. The mentor selected the appropriate calyx. Depending on the surgeon’s preference, either a 30 Fr Amplatz dilator set (Actomed, Ankara, Turkey) or a 30 Fr NephroMax^®^ Balloon dilator (Boston Scientific, Marlborough, MA, USA) was used. Continuous fluoroscopy was avoided throughout the surgery, and pulse fluoroscopy (intermittent use) was used in each surgery to minimize radiation exposure to the surgeon and the patient. After gaining access to the collecting system, stones in the calyx structures were fragmented using a pneumatic lithotriptor (Vibrolith, Elmed, Turkey) and a 26 Fr rigid nephroscope (Karl Storz, Tuttleen, Germany) before being removed with forceps. Postoperatively, a 4.7 Fr 28 cm DJ stent was placed in an antegrade manner in the supine position in all patients who underwent supine PCNL and in the lithotomy position in all patients who underwent prone PCNL, and a 14 Fr nephrostomy tube was placed according to the surgeon’s preference. The stone-free status of the patients was checked by kidney ureter bladder radiography (KUB), ultrasonography (USG), or CT one month after surgery. If the patient was considered stone-free postoperatively, they were evaluated using KUB or USG, according to the mentor’s preference, to avoid additional radiation exposure. If clinically significant residuals were suspected or additional treatment was needed, the patient was evaluated using a CT scan. The nephrostomy tube, if any, was clamped on the first day after surgery and removed 6 h later if there was no flank pain or leaking around the nephrostomy tube. A hemogram was obtained from each patient on the first day after surgery.

This study was reviewed and approved by the Medical Ethics Committee of Gazi University, School of Medicine, on 3 April 2023 (approval number: 284).

### Statistical Analysis

Statistical analysis was carried out using the SPSS software (Statistical Package for the Social Science, version 23, Armonk, NY, USA). Categorical data across groups were compared by the Chi-square test and continuous variables were compared by the Kruskal–Wallis test. Statistical significance was accepted with a *p*-value less than 0.05.

## 3. Results

The average age of the patients was 55.5 ± 15.44 years in the supine PCNL group and 49 ± 13.08 years in the prone PCNL group. There was no statistically significant difference between the two groups (*p* > 0.05). The median HU values were 1546 (550–2800) and 1113 (502–2400) in the supine and prone PCNL groups, respectively. There was a statistically significant difference in HU values between the two groups (*p* < 0.001). There was no statistically significant difference between the two groups regarding gender, BMI, side, stone location, stone size, stone-free rate, and ES replacement (*p* > 0.05). The mean operation time was 94.6 ± 9.8 min in the supine PCNL group and 129.9 ± 20.3 min in the prone PCNL group (*p* < 0.001). Fluoroscopy time was statistically significantly lower in the supine PCNL group compared to the prone PCNL group. The median values of fluoroscopy time were 31 (10–89) seconds and 48 (23–156) seconds for the supine and prone groups, respectively (*p* = 0.001) (Table 1).

There was no significant difference in stone-free rates among the five subgroups in the supine and prone PCNL groups (*p* > 0.05). In the supine PCNL group, the stone-free rate reached 90% after the 20th case, while in the prone PCNL group, the highest stone-free rate, 80%, was reached after the 30th case. A statistically significant difference in operation time was observed among the supine and prone PCNL subgroups (*p* = 0.037 and *p* = 0.002, respectively). After the first 10 cases in both the supine PCNL and prone PCNL groups, a rapid decrease in operation time was observed (Figure 1).

A statistically significant difference was observed among supine PCNL subgroups in terms of fluoroscopy time, but no significant difference was observed among prone PCNL subgroups (*p* = 0.005 and *p* = 0.227, respectively) (Table 2).

No significant difference was found in terms of hemoglobin (Hb) decrease and ES replacement among the two position subgroups (*p* > 0.05). However, in the supine PCNL group, 10% of the patients underwent erythrocyte suspension (ES) replacement in the 11th–20th case group; in the prone PCNL group, this rate was reached in the 31st–40th case group. A significant difference was observed among prone PCNL subgroups in terms of stone size (*p* = 0.03) (Table 2).

## 4. Discussion

Our study aimed to determine the preferable position in training clinics by analyzing its impact on PCNL techniques and outcomes during the learning phase. In our clinic, a reference center in our country in terms of urinary system stone treatment, PCNL is performed in both supine and prone positions by different surgeons, and these operations have an essential place in resident training.

One of the most important goals of the PCNL operation is to achieve a high stone-free rate (SFR). Looking at the results of studies examining the effects of positions on the SFR, in two prospective randomized studies, Choudhury et al. and Seleem et al. compared supine and prone PCNL, reporting no significant difference in the SFR between the two methods [10,11]. In two separate meta-analyses examining 7733 and 1290 PCNL cases, no difference was seen between the supine PCNL and prone PCNL groups regarding the stone-free rate [12,13]. On the contrary, in the meta-analysis of Yuan et al., they found that the supine position was associated with a higher rate of stone-free status [14]. In the Birowo et al. meta-analysis, the SFR was higher in prone PCNL [15]. In our study, surgeons in the learning phase achieved a higher stone-free rate in the supine PCNL group, though this difference was not statistically significant, likely due to the limited sample size. It would not be surprising to expect a higher SFR in supine PCNL due to the spontaneous retrieval effect. In addition, it should be considered that in our study, the HU was higher in the supine PCNL group, and possibly that harder stones were treated. In the supine mini-PCNL study by Sahan et al., which divided 15 patients into groups, no difference was observed between the groups regarding stone-freeness, but the stone-free rate increased to 93% in the fourth group [16]. In our study, the surgeon in the learning phase reached a high stone-free rate in the supine PCNL group earlier than the surgeon in the prone PCNL group (after the 20th patient). However, at the end of the training, in a clinic with high patient numbers, a high SFR can be achieved in both positions.

In the study conducted by Chapagain et al. in which the effect of the two positions on the operation time was compared, it was observed that the operation time was shorter in supine PCNL than in prone PCNL [17]. This may be because positioning the patient takes less time in supine PCNL compared to prone PCNL. However, in the study of Mulay et al., the times from puncture to the end of the surgery were compared, and it was observed that the surgery time was shorter in the supine position [18]. This may be due to the spontaneous drainage of stone fragments in supine PCNL. In the study by Melo et al., which evaluated PCNL outcomes in the prone position and three different supine positions (complete supine, original Valdivia, and Galdakao-modified Valdivia), it was found that the operation and fluoroscopy times were shorter in the complete supine position. The authors suggested that this might be due to the surgeons having more experience with this position [19]. In the meta-analysis of Falahatkar et al., no significant difference was found in terms of operation time in both positions [12]. In our study, the operation time was significantly lower in the supine PCNL group than in the prone PCNL group. The most important reason for this may be that supine PCNL has the advantage of spontaneous stone removal, and no time is wasted on small fragments. Additionally, in the prone PCNL group, position change may occur during the ureteral catheter and ureteral stent assembly. In the prone PCNL group, the median HU value was lower. Therefore, it could be expected that the stones will be less hard and the lithotripsy time will be shorter, potentially reducing the operation time. However, it can be argued that the advantages of spontaneous stone removal, not losing time on small fragments, and not having to change positions outweigh the disadvantages caused by harder stones.

Studies defining the learning curve in PCNL typically evaluate operation time, fluoroscopy time, stone-free rates, and complication rates [20,21]. In their study, Tanrıverdi et al. examined the operating times for the learning curve in PCNL performed in the prone position. They showed that the operating times plateaued after the first 60 cases [21]. In the prone PCNL study of Ziaee et al., a plateau was observed in the operation time after 45 cases [22]. The prone PCNL study of Allen et al. showed that the novice surgeon had the shortest operating time in the 46th–60th case group [20]. In the supine mini-PCNL study by Sahan et al., which divided the study population of 15 patients into groups, although there was no significant difference found between the groups in terms of operation time, it reached the lowest level in the fourth group [16]. In our study, consistent with the literature, the operation time in the prone group reached the shortest time after the 40th patient. However, in the supine PCNL group, after the first 10 cases, a significant shortening was observed in the operation time and a plateau began.

PCNL can be performed using fluoroscopy or ultrasound. When using fluoroscopy, the surgeon and the patient are exposed to radiation. The short duration of fluoroscopy reduces this exposure. In the study of Zampini et al., it was observed that fluoroscopy was used for a more extended period in supine PCNL [23]. In contrast, our study found that surgeons in the learning phase performed supine PCNL with shorter fluoroscopy times. Considering that a significant portion of the fluoroscopy time in PCNL comprises access and dilation, the access time may be shorter in the supine PCNL group. The reason for this may be that the patient’s position is similar to the tomography images examined before the operation and the fluoroscopy images during the operation may facilitate three-dimensional thinking.

Intraoperative bleeding is one of the most common complications of PCNL operations, and its severity can be assessed by ES replacement. In our study, there was no difference in ES replacement rates between the supine and prone PCNL groups among surgeons in the learning phase. Falahatkar et al. reported in their study that blood transfusion rates in supine PCNL were lower than in prone PCNL [12]. Karami et al., who compared PCNL operations performed in the prone, supine, and flank positions, did not report any difference in blood transfusion between all three positions in their study [24]. Walick et al. reported that, although it is not very common during the operation in prone PCNL, vision loss may occur due to increased intraocular pressure [6]. Reducing the risk of circulatory and respiratory disorders and allowing for ureterorenoscopy (URS) during the operation are other advantages of adopting the supine position [25]. The small number of patients in our study, the absence of rare complications, and the exclusion of endoscopic combined intrarenal surgeries prevented us from performing these examinations for novice surgeons.

To the best of our knowledge, our study may be the first study in the literature to examine the effect of supine and prone positions on the PCNL learning curve. The results of this study may influence the choice of positions for PCNL in training clinics. Operation and fluoroscopy results can give an idea about which position is preferable. However, our study has significant limitations. The retrospective design of our study is one of the most essential bias steps. In addition, the limited number of patients due to the duration of the chief residency and the creation of two groups of 50 patients may have prevented the analyses from obtaining solid results. Additionally, not using standardized equipment for dilatation may have affected the fluoroscopy time. The CT images were accessed through the national health database (e-nabız) of patients whose preoperative CT scans were taken at external centers. Since the 3D volume measurement feature is unavailable in this database, the stone size was measured based on the largest diameter. The evaluation of stone size in one dimension may not be an objective indicator of stone burden and is one of the most important biases of this study. The two groups differed in demographic characteristics, such as the HU, which may affect surgery outcomes. Placement of the ureteral stent in the lithotomy position in the prone PCNL group, as opposed to in an antegrade manner in the supine PCNL group, may have affected surgery duration. The use of different imaging methods for stone-free evaluation reduces the reliability in terms of SFRs. Although the time taken to successfully access the urinary system is one of the fundamental steps in PCNL, we could not include this duration in the present study because the access time was not available for some cases in our database. However, the fact that fluoroscopy is predominantly used during the access stage of PCNL could suggest that this time might also be shorter.

## 5. Conclusions

Based on our study findings, the supine position offers superior stone-free rates, shorter operation times, and reduced fluoroscopy durations during the PCNL learning phase. It can also be inferred that reaching the plateau in terms of duration was achieved earlier in this group. However, prospective randomized studies with high patient numbers can obtain more robust inferences.

## Figures and Tables

**Figure 1 medicina-60-01292-f001:**
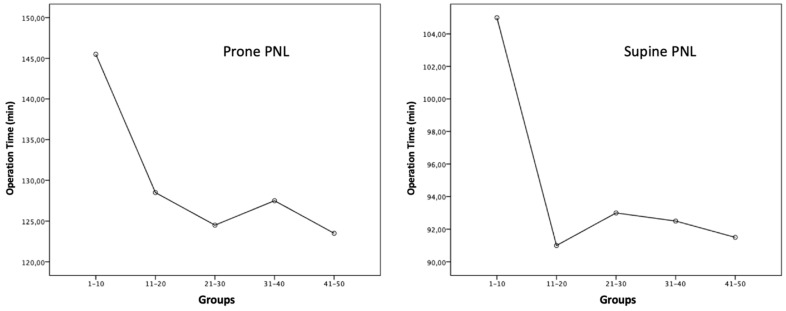
Operation time comparison between supine and prone PCNL.

**Table 1 medicina-60-01292-t001:** Demographic and clinical characteristics of supine and prone PCNL groups.

Characteristics	Supine PCNL (*n* = 50)	Prone PCNL (*n* = 50)	*p*
Age (years) (mean ± SD)	55.5 ± 15.44	49 ± 13.08	0.275
Sex			0.065
Male n (%)	35 (70%)	26 (52%)	
Female n (%)	15 (30%)	24 (48%)	
BMI (kg/m^2^) (median) (min-max)	26.4 (19–39)	26.1 (18–40)	0.997
Side (right/left)			0.841
Right n (%)	27 (54%)	28 (56%)	
Left n (%)	23 (46%)	22 (44%)	
Stone location			0.959
Pelvis n (%)	28 (56%)	24 (48%)	
Lower calyx n (%)	15 (30%)	12 (24%)	
Middle calyx n (%)	8 (16%)	13 (26%)	
Upper calyx n (%)	12 (24%)	8 (16%)	
Ureteropelvic junction n (%)	5 (10%)	6 (12%)	
Staghorn n (%)	9 (18%)	8 (16%)	
Stone size (mm) (median) (min-max)	20 (8–46)	24.5 (10–49)	0.133
Hounsfield unit (median) (min-max)	1546 (550–2800)	1113 (502–2400)	**<0.001**
Stone-free rate n (%)	40 (80%)	32 (64%)	0.075
ES replacement n (%)	7 (14%)	9 (18%)	0.585
Operation time (min) (mean ± STD)	94.6 ± 9.8	129.9 ± 20.3	**<0.001**
Fluoroscopy time (s) (median) (min-max)	31 (10–89)	48 (23–156)	**0.001**

BMI: body mass index; ES: erythrocyte suspension. *p* < 0.05: statistically significant.

**Table 2 medicina-60-01292-t002:** Comparison of subgroups undergoing supine and prone PCNL.

**Supine PCNL**						
Groups	1–10	11–20	21–30	31–40	41–50	*p*
Stone-free rate n (%)	6 (60%)	7 (70%)	9 (90%)	9 (90%)	9 (90%)	0.464
Operation time (min) (mean ± STD)	105 ± 11.3	91 ± 10.7	93 ± 7.8	92.5 ± 7.5	91.5 ± 2.4	**0.037**
Fluoroscopy time (sec) (median) (min-max)	48.5 (24–89)	32.5 (21–89)	34.5 (10–85)	24.5 (11–33)	22.5 (15–42)	**0.005**
ES replacement n (%)	3 (30%)	1 (10%)	1 (10%)	1 (10%)	1 (10%)	0.643
Hb drop (gr/dL) (median) (min-max)	1.45 (0.2–5.5)	1.25 (−0.4–3.7)	0.95 (−0.6–2.2)	1.2 (−0.1–3.2)	1.4 (0.6–2.2)	0.355
Stone size (mm) (median) (min-max)	24.5 (15–43)	20 (11–29)	15.5 (12–35)	21.5 (8–45)	22.5 (12–46)	0.571
**Prone PCNL**						
Groups	1–10	11–20	21–30	31–40	41–50	*p*
Stone-free rate n (%)	5 (50%)	5 (50%)	6 (60%)	8 (80%)	8 (80%)	0.451
Operation time (min) (mean, SD) (min-max)	145.5 ± 5.5 (135–150)	128.5 ± 22 (90–150)	124.5 ± 20.2 (90–145)	127.5 ± 22.2 (80–160)	123.5 ± 21.6 (90–150)	**0.002**
Fluoroscopy time (sec) (median) (min-max)	54.5 (23–156)	57 (38–92)	44 (25–77)	32.5 (23–76)	36.5 (27–77)	0.227
ES replacement n (%)	3 (30%)	2 (20%)	2 (20%)	1 (10%)	1 (10%)	0.776
Hb drop (gr/dL) (median) (min-max)	1.2 (−0.5–2.2)	1.15 (−0.1–3.5)	1.7 (0.7–2.2)	1 (−0.3–3.4)	1.25 (−0.2–2.6)	0.821
Stone size (mm) (median) (min-max)	27 (19–38)	31 (17–42)	25 (10–49)	24.5 (13–40)	17 (13–30)	**0.03**

ES: erythrocyte suspension; Hb: hemoglobin. *p* < 0.05: statistically significant.

## Data Availability

The data that support the findings of this study are available from the corresponding author upon reasonable request.

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
