# Peer review of "Which Position for Novice Surgeons? Effect of Supine and Prone Positions on Percutaneous Nephrolithotomy Learning Curve"

_medicina, 2024, doi:10.3390/medicina60081292_

Round 1
Reviewer 1 Report
Comments and Suggestions for Authors
Any particular reason why this was done in a retrospective fashion? if done prospectively and randomised, would have been more convincing data.
note that prone patients are not stented antegrade? need to position back to lithotomy - hence leading to increased operative time.
Also looking at table 2, the stone size (median ) appears to be larger for the patients in prone group.
Comments on the Quality of English LanguageAble to understand.
Author Response
Dear Reviewer 1;
First of all, thank you for your interest in the study and your comments. We tried to respond to all your comments and revised the article according to your views. We have no doubt that the scientific value of the study has increased with your suggestions.
Comments 1: Any particular reason why this was done in a retrospective fashion? if done prospectively and randomized, would have been more convincing data.
Response 1: In the study, one of the mentors who trained the chief assistants performed PCNL in the supine position, while the other performed it in the prone position. You are absolutely correct in your comment. A prospective design of the study would have made our results more reliable. However, the mentor performing PNL in the prone position has started performing it in the supine position for the last six months. This makes it impossible for our clinic to design studies prospectively that require a comparison between Supine and Prone PCNL. The retrospective design of the study has been noted as one of the most essential sources of bias in the limitations. It has been highlighted in red in the text.
Comments 2: Note that prone patients are not stented antegrade? need to position back to lithotomy - hence leading to increased operative time.
Response 2: The mentor performing PNL in the prone position routinely places the double-J stent in the lithotomy position in a retrograde manner, with the concern that if the distal end of the stent stays intraurethral or intraureteral, it would be difficult to correct the situation or it would require returning to the lithotomy position for correction.
You are absolutely right that it could affect the operation time. We have included this point in the limitations. And it has been highlighted in red in the text.
Comments 3: Also looking at table 2, the stone size (median) appears to be larger for the patients in prone group.
Response 3: The median values in Table 2 could be higher than those in Table 1 due to the data being non-parametric. In Table 2, the median stone sizes for the first 3 subgroups in the Prone PNL group are higher than the overall group's median, while the last 2 subgroups have lower median values. The result remains the same despite reevaluating the data as you suggested. The median value for the Prone PCNL group was again calculated as 24.5 using SPSS version 23.
Thank you for your suggestions to make the article more understandable for the readers and correct the mistakes in the design of the article. We hope we were able to adapt our article to your recommendations. The corrections you suggested have been highlighted in red in the revised version of the manuscript.
Best Regards
Reviewer 2 Report
Comments and Suggestions for Authors
Dear authors, I read your work with great interest. The supine and prone position is a focus of endourology treatment. I find a nice and well-orientated paper. Nevertheless, it would be highly appreciated if some data could be explicit, for example: the time to successfully access the urinary system, as it is one of the fundamental steps in PCNL. I am also concerned about the stone size difference that was significant in the prone position, could the analysis of time be weighted against stone size?
Author Response
Dear Reviewer 2;
First of all, thank you for your interest in the study and your comments. We tried to respond to all your comments and revised the article according to your views. We have no doubt that the scientific value of the study has increased with your suggestions.
Comments 1: it would be highly appreciated if some data could be explicit, for example: the time to successfully access the urinary system, as it is one of the fundamental steps in PCNL.
Response 1: You are absolutely right that the time to successfully access the urinary system is a fundamental step in PCNL. However, the retrospective nature of the study and the absence of access time data for some cases in our database prevented us from evaluating this parameter. We have added this point to the limitations section based on your comment.
Comments 2: I am also concerned about the stone size difference that was significant in the prone position, could the analysis of time be weighted against stone size?
Response 2: As mentioned in the discussion section, we believe the shorter operation time is primarily due to the lack of need for intraoperative position changes and the similarity between the intraoperative fluoroscopy images of the urinary system and the preoperative CT images. Weighting the operation time by stone size would require a change in the study design. However, we must say that your suggestion has inspired us to design a new study.
Thank you for your suggestions to make the article more understandable for the readers and correct the mistakes in the design of the article. We hope we were able to adapt our article to your recommendations. The corrections you suggested have been highlighted in green in the revised version of the manuscript.
Best Regards
Reviewer 3 Report
Comments and Suggestions for Authors
Dear Authors,
The manuscript submitted to Medicina is interesting. Percutaneous nephrolithotomy (PCNL) was developed for patients in the prone position at the start of the 1980s. The supine position certainly offers some advantages over prone PCNL in terms of anesthesiological management. No consensus exists on the best positioning in PCNL, nor is it clear which variation of the supine position is best. The pros of supine PCNL outweigh the cons, which will soon be appreciated by most urologists worldwide, especially young urologists.
Here are my comments:
Title - I suggest changing to Supine or Prone
l. 8 PCNL is the most frequently used abbreviation
l. 17 and 18 seconds??? maybe minute
In the introduction section, please include the indications of PNL according to the guidelines
l. 34 please change providing to provide
Materials and Methods - Were there any exclusion criteria? Did the authors include patients with previous surgery for renal lithiasis?
l. 48 It is important to specify whether the surgeons performed any part of PNL in the residency, as well as their number and position.
l. 49 Please define assistantship and chief assistants
l. 53 please change to PNL cases
The authors should include the volume of the stone. Otherwise, there is a lot of bias, not only the largest diameter.
l. 57 How do the authors evaluate stone-free and when?
l 62. How do you obtain a sterile urine culture of a patient with renal lithiasis if infected? In most cases, it is impossible
l. 67 The puncture was only to the inferior calyx; please specify
l. 77-78 Did the authors use this investigation in all patients? Why do they perform KUB, USG, and CT?
l. 83 Why did the authors obtain the approval after completing the study?
l. 100 The fluoroscopy time is too low. Please try to explain
Table 1 - it is strange that the operation time is higher in the Prone group and the Hounsfield Unite is low. Please include a comment in the Discussion section
l. 127. Please include the total number of PNLs performed in your clinic in the last five years, both supine and prone.
Table 2 should be included in the Results section. The indication for PNL is renal lithiasis over 2 cm or lower calyx over 1.5 cm. Why did the authors perform PNL for lithiasis under 1.5 cm?
The References list should be updated. There are only five from the last five years.
Please include the following article
doi: 10.1590/S1677-5538.IBJU.2018.0191
Comments on the Quality of English Language
I suggest a native English speaker should read the paper.
Author Response
Dear Reviewer 3;
First of all, thank you for your interest in the study and your comments. We tried to respond to all your comments and revised the article according to your views. We have no doubt that the scientific value of the study has increased with your suggestions.
Comments 1: Title - I suggest changing to Supine or Prone
Response 1: The title you suggested could indeed have been more compelling for consideration of the paper. However, even though the title can be changed in the manuscript, the initial title information entered into the system cannot be altered. To avoid any confusion during the publication process, we were unfortunately unable to change the title.
Comments 2: PCNL is the most frequently used abbreviation
Response 2: You are correct that the abbreviation PCNL is more commonly used in the literature. We used the abbreviation PNL because it is the standard in our department. Based on your suggestion, we have adopted the abbreviation PCNL throughout the entire paper to refer to percutaneous nephrolithotomy.
Comments 3: 17 and 18 seconds??? maybe minute
Response 3: Thanks for your attention. Operation time has been corrected as minutes instead of seconds.
Comments 4: In the introduction section, please include the indications of PNL according to the guidelines
Response 4: Based on your suggestion, we have added the PCNL indication recommended by the guidelines.
Comments 5: please change providing to provide
Response 5: The necessary correction has been made according to your suggestion.
Comments 6: Materials and Methods - Were there any exclusion criteria? Did the authors include patients with previous surgery for renal lithiasis?
Response 6: Consecutive prone and supine PCNL data performed by two separate chief residents as primary surgeons under the control of a mentor were examined. Patients who underwent endoscopic combined intrarenal surgery (ECIRS) were excluded from the study. (lines 46-47 and lines 67-68). Only cases where the chief residents were the primary surgeons were examined. The exclusion criteria were limited because, in challenging or complicated cases, the chief residents were not the primary surgeons.
Comments 7: It is important to specify whether the surgeons performed any part of PNL in the residency, as well as their number and position.
Response 7: Actually, we had mentioned this part in the materials and methods section, but we have elaborated on it further based on your suggestion.
Comments 8: Please define assistantship and chief assistants
Response 8: Based on your suggestion, we have defined the terms 'residency' and 'chief residency' in the materials and methods section. (lines 50-54, 57-58)
Comments 9: please change to PNL cases
Response 9: The necessary correction has been made according to your suggestion.
Comments 10: The authors should include the volume of the stone. Otherwise, there is a lot of bias, not only the largest diameter.
Response 10: You are absolutely right that measuring the stone size in only one dimension introduces a significant bias. We had noted this in the limitations, but we have emphasized it further based on your suggestion. CT scans taken at external centers are viewed through the national health database (e-nabız). Unfortunately, this database does not have a 3D volume measurement feature. We have also added this information to the limitations section. (lines 62-63, 245-250)
Comments 11: How do the authors evaluate stone-free and when?
Response 11: Stone-free status of the patients was checked by kidney ureter bladder radiography (KUB), ultrasonography (USG), or CT one month after surgery. We had mentioned this information in the materials and methods section. The use of different imaging methods may have impacted objective evaluation. We addressed this issue in the limitations section as well. (lines 90-91, 253-254)
Comments 12: How do you obtain a sterile urine culture of a patient with renal lithiasis if infected? In most cases, it is impossible
Response 12: If the preoperative urine culture is positive, the patient is treated with the appropriate antibiotics based on the antibiogram results as an outpatient or inpatient. As you mentioned, achieving sterile urine can sometimes be impossible in certain patients. These patients are operated on under antibiotic suppression with the approval of the Infectious Diseases Department. However, in our clinic, these cases are considered challenging and complicated and are operated on by a mentor, not by a chief resident. Consequently, they are not included in the cases analyzed in this study. Based on your suggestion, we have added to the materials and methods section that patients with positive urine cultures received appropriate antibiotic treatment to achieve sterile urine before the operation. (lines 69-71)
Comments 13: The puncture was only to the inferior calyx; please specify
Response 13: Access was obtained from the most suitable calyx, not only the inferior calyx to ensure adequate lithotripsy and stone extraction. The mentor selected the appropriate calyx. Based on your suggestion, this statement has been added to the materials and methods section. (lines 78-79)
Comments 14: Did the authors use this investigation in all patients? Why do they perform KUB, USG, and CT?
Response 14: All patients are routinely evaluated with an imaging method one month after the operation. The ureteral stent is removed 1-month post-PCNL, and if clinically significant residual fragments are seen on imaging, additional treatment is planned for one month later. If the patient is considered stone-free after PCNL, a KUB or ultrasound (USG) is performed, depending on the mentor's preference, to avoid additional radiation exposure. If residual stones are suspected and further treatment is necessary, the patient is re-evaluated with a CT scan. The evaluation of patients using imaging has been detailed further based on your suggestion. (lines 91-94)
Comments 15: Why did the authors obtain the approval after completing the study?
Response 15: Since the study was designed retrospectively, the data used were from the period (August 2021 and January 2023) before the ethical committee approval date. The study design was finalized after obtaining ethical committee approval on April 3, 2023. Patient consent was also obtained after the study design was completed.
Comments 16: The fluoroscopy time is too low. Please try to explain
Response 16: In our clinic, to minimize radiation exposure to both the surgeon and the patient during PCNL, we conventionally avoid the use of continuous fluoroscopy. Instead, we prefer the intermittent use (pulse-fluoroscopy), which significantly reduces the fluoroscopy time. Based on your comment, we have added the necessary information to the materials and methods section. (lines 81-83)
Comments 17: it is strange that the operation time is higher in the Prone group and the Hounsfield Unite is low. Please include a comment in the Discussion section
Response 17: As you mentioned, the median HU value was lower in the Prone PCNL group. Since the study was retrospective and only the 50 cases performed by the chief residents were selected, we could not control for this difference. The lower HU value suggests that the stones are less hard, which could potentially reduce lithotripsy and operation time. However, we believe that the advantages of spontaneous stone removal, not losing time with small fragments, and not needing to change patient positions outweigh the disadvantages of dealing with harder stones.
Based on your comment, we discussed this issue in the relevant section. (lines 196-200)
Comments 18: Table 2 should be included in the Results section. The indication for PNL is renal lithiasis over 2 cm or lower calyx over 1.5 cm. Why did the authors perform PNL for lithiasis under 1.5 cm?
Response 18: The location of Table2 has been changed based on your suggestion.
Your concern is valid; PCNL may be considered an overtreatment for stones under 15mm.
In our data set, in the Supine PNL group, there was one patient with an 8mm stone, and in the Prone PNL group, there was one patient with a 10mm stone. These stones were located in the lower calyx, and due to the narrow pelvis-calyx angles, they could not be accessed with a flexible ureteroscope. For stones under 15mm, the mentor decided to perform PCNL based on clinical evaluation (such as narrow pelvis-calyx angles or low ureter calibration, etc.). According to the EAU Guidelines, while PCNL is not the first choice for kidney stones of all sizes, it is considered a treatment option when clinically necessary.
EAU Guidelines/Non-oncology guidelines/Urolithiasis Edn. presented at the EAU Annual Congress Paris 2024. ISBN 978-94-92671-23-3.
Comments 19: The References list should be updated. There are only five from the last five years. Please include the following article doi: 10.1590/S1677-5538.IBJU.2018.0191
Response 19: A reference related to your suggestion has been added. In the final version of the paper, 10 out of the 25 references are from the last 5 years. The reference numbers are 2, 10, 11, 13, 15, 16, 17, 18, 19, and 23.
Thank you for your suggestions to make the article more understandable for the readers and correct the mistakes in the design of the article. We hope we were able to adapt our article to your recommendations. The corrections you suggested have been highlighted in yellow in the revised version of the manuscript.
Best Regards
Round 2
Reviewer 3 Report
Comments and Suggestions for Authors
Dear Authors,
Congratulations on your work. The manuscript has been improved.
I still have a few suggestions:
l. 29 and over 1.5 cm in the lower calyx
Materials and Methods
Did the authors include patients with previous surgery for renal lithiasis?
l. 70 Please specify the number of days
Comments on the Quality of English LanguageI suggested the manuscript to be read by a native English speaker.
Author Response
Dear Reviewer,
Thank you very much for your interest in our manuscript and for your valuable contributions, which reflect your high academic vision and are beneficial for the reader. We have made the suggested corrections and highlighted them in yellow. Additionally, we had the manuscript reviewed by a native English speaker, who suggested a few minor revisions.
Comments 1: and over 1.5 cm in the lower calyx
Response 1: The phrase 'and over 1.5 cm in the lower calyx' was added.
Comments 2: Did the authors include patients with previous surgery for renal lithiasis?
Response 2: The condition of not having undergone surgery before was not required for inclusion in the study. Patients with a history of previous surgeries were also included in the study. (lines 68-69)
Comments 3: Did the authors include patients with previous surgery for renal lithiasis?
Response 3: The duration of antibiotic treatment ranged from 7 to 10 days. This information has been added to the relevant section based on your suggestion.
We hope that the manuscript now meets the academic standards you expected.
Best regards